# Tracking excited state decay mechanisms of pyrimidine nucleosides in real time

Rocío Borrego-Varillas [1,7], Artur Nenov [2,7], Piotr Kabaciński [3], Irene Conti[2], Lucia Ganzer[3], Aurelio Oriana[3], Vishal Kumar Jaiswal [2], Ines Delfino[4], Oliver Weingart[5], Cristian Manzoni [1], Ivan Rivalta [2,6], Marco Garavelli [2✉] & Giulio Cerullo [1,3✉]

DNA owes its remarkable photostability to its building blocks—the nucleosides—that efficiently dissipate the energy acquired upon ultraviolet light absorption. The mechanism occurring on a sub-picosecond time scale has been a matter of intense debate. Here we combine sub-30-fs transient absorption spectroscopy experiments with broad spectral coverage and state-of-the-art mixed quantum-classical dynamics with spectral signal simulations to resolve the early steps of the deactivation mechanisms of uridine (Urd) and 5-methyluridine (5mUrd) in aqueous solution. We track the wave packet motion from the Franck-Condon region to the conical intersections (CIs) with the ground state and observe spectral signatures of excited-state vibrational modes. 5mUrd exhibits an order of magnitude longer lifetime with respect to Urd due to the solvent reorganization needed to facilitate bulky methyl group motions leading to the CI. This activates potentially lesion-inducing dynamics such as ring opening. Involvement of the $^1n\pi^\star$ state is found to be negligible.

[1] IFN-CNR, Piazza Leonardo da Vinci 32, Milano, Italy. [2] Dipartimento di Chimica Industriale, Università degli Studi di Bologna, Viale del Risorgimento 4, Bologna, Italy. [3] Dipartimento di Fisica, Politecnico di Milano, Piazza Leonardo da Vinci 32, Milano, Italy. [4] Dipartimento di Scienze Ecologiche e Biologiche, Università della Tuscia, Via San Camillo de Lellis, snc, Viterbo, Italy. [5] Institut für Theoretische Chemie und Computerchemie, Heinrich Heine Universität Düsseldorf, Universitätsstrasse 1, Düsseldorf, Germany. [6] Université de Lyon, École Normale Supérieure de Lyon, CNRS UMR 5182, Laboratoire de Chimie, 46 allée d'Italie, F69364 Lyon, France. [7] These authors contributed equally: Rocío Borrego-Varillas, Artur Nenov. ✉email: marco.garavelli@unibo.it; giulio.cerullo@polimi.it

Ultraviolet (UV) radiation can induce significant photo-damage to biomolecules. This is particularly true for nucleosides, the building blocks of DNA, which display strong absorption bands in the UV. Due to the high energy of the UV photons, the excess electronic energy acquired by the molecule could initiate a chain of photochemical reactions ultimately altering the structure of the base sequence[1–4]. However, in most cases photoexcitations in DNA do not trigger reactions: the excess of electronic energy is dissipated on ultrafast timescales ranging from sub-picosecond[5–10] in isolated nucleosides to few hundreds of picoseconds in base pairs, single strands and double stranded DNA, resulting in effective photoprotection mechanisms[11–14].

It is now generally accepted that conical intersections (CIs) play a crucial role in the deactivation pathways of nucleosides[15] and are responsible for the ultrafast internal conversion (IC) to the ground state. However, no consensus has been reached yet on the decay mechanisms. In particular, in pyrimidines, the time scale of IC and the involvement of intermediate dark states are still disputed. Experimental transient absorption (TA), photo-electron spectroscopy and fluorescence up-conversion measurements on pyrimidines and their nucleosides in aqueous solution agree on the early fate of the initially excited ($^1\pi\pi^*$) singlet state, that decays with one or two sub-picosecond components[6,8,16–23] and a further component of several picoseconds[6–8,16–21,24–27]. The sub-100 fs time constant has been so far assigned to the departure from the bright state for both nucleosides, but there is still disagreement on whether the decay is directly to the ground state[20,22] or, alternatively, to the $^1n\pi^*$ state[17,24]. Even more scenarios have been proposed regarding the long component, including ground state vibrational cooling, intersystem crossing (ISC) to triplet states or IC of dark states to the ground state. A review of the different hypotheses and a table with the experimental time constants are provided in Supplementary Note 1. Thus, key questions regarding the assignment of the experimental decay times, the involvement of dark states, and the molecular processes responsible for the higher reactivity of thymine, remain open. Furthermore, the role of the solvent and how it affects the dynamics is still to be explored.

Experimental and theoretical bottlenecks have so far hindered the unambiguous definition of the molecular mechanisms underlying pyrimidine excited state deactivation. From an experimental point of view, the lack of temporal resolution and the limited spectral coverage of the probe pulses have prevented an accurate tracking of dynamics taking place on the sub-100-fs timescale. From a theoretical point of view, simulations of excited states dynamics have been mostly carried out in semi-classical fashion, sacrificing accuracy either by neglecting the environmental effects (i.e., assuming vacuum environment) or by adopting quantum-mechanical (QM) treatments of the electronic structures, with limited description of electronic correlations. Moreover, UV–Vis transient absorption spectroscopy simulations from first principles translating the molecular dynamics into experimentally observable spectral features have been missing so far.

Here we perform TA spectroscopy in the pyrimidine nucleosides uridine (Urd) and 5-methyluridine (5mUrd) in aqueous solution with ~30-fs temporal resolution and full coverage of the 1.9–4.2 eV probe spectral range. Combining the experimental results with simulations based on mixed quantum-classical dynamics within a hybrid quantum mechanics/molecular mechanics (QM/MM) framework explicitly including environmental effects and electronic correlation, we are able to reproduce the experiments and assign the observed spectral signals and the corresponding time constants to the specific pathways responsible for the decay.

We resolve experimentally the spectral dynamics on sub-100 fs timescale in a broad UV-visible window, elusive up to now due to

a low temporal and/or spectral resolution or to the presence of strong coherent artifacts, and report real-time observation of vibrational signatures of the coherent wave packet motion in a DNA nucleoside towards the CI seam. Our results show that, in both Urd and 5mUrd, IC through ring puckering is the main decay mechanism to the ground state, but 5mUrd takes an order of magnitude longer ($\approx$1 ps versus $\approx$100 fs) to reach the crossing region. At odds with previous interpretations, our study shows that a decay to the ground state (GS) on the time scale of few hundred femtoseconds does not occur in 5mUrd due to a solvent-induced dynamic barrier associated with the bulky methyl group involved in the ring puckering. This effect adds to the already larger (with respect to Urd) inertia of 5mUrd due to the heavier methyl group, as recognized in earlier studies[10]. This leads to a dynamic trapping of the 5mUrd population in the $^1\pi\pi^*$ state for which we observe stimulated emission (SE) lasting for picoseconds as a compelling spectroscopic signature. These results provide a novel view of the decay process that calls for a key contribution of solvent reorganization, thereby disentangling two contributions: a static (destabilization of the $n\pi^*$ state with respect to the $\pi\pi^*$ state) and a dynamic (slowing down ultrafast bulky geometrical deformations) one. Due to the longer ES lifetime, secondary deactivation channels to the GS such as oxygen ($O_8$) out-of-plane deformation and $N_1$-$C_2$ ring-opening become accessible in 5mUrd. Remarkably, despite spending almost an order of magnitude shorter time in the excited state (ES), Urd is more prone to involve the $^1n\pi^*$ state in the non-adiabatic decay, although as a minor channel.

## Results and Discussion

**Transient absorption spectra of pyrimidines.** Figure 1c shows TA spectra of Urd in phosphate buffer solution, recorded with sub-30-fs temporal resolution[28,29] following photoexcitation at 4.5 eV. At early times we observe an intense SE band in the UV, covering the 3.00–4.00 eV spectral range, as well as a

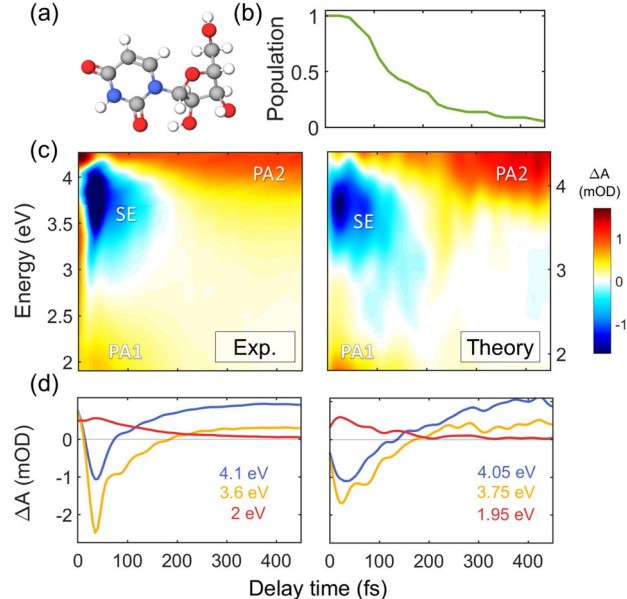

**Fig. 1 Excited state dynamics of uridine. a** Molecular structure of Urd. **b** Time-dependent excited state population calculated by QM/MM simulations. **c** Experimental (left) and computed (right) TA maps of Urd. Two photo-induced absorption bands (labeled PA1 and PA2) and a stimulated emission (SE) band can be identified. **d** TA dynamics at selected probe photon energies; signal intensity reported as differential optical density ΔA in mOD units. Source data are provided as a Source Data file.

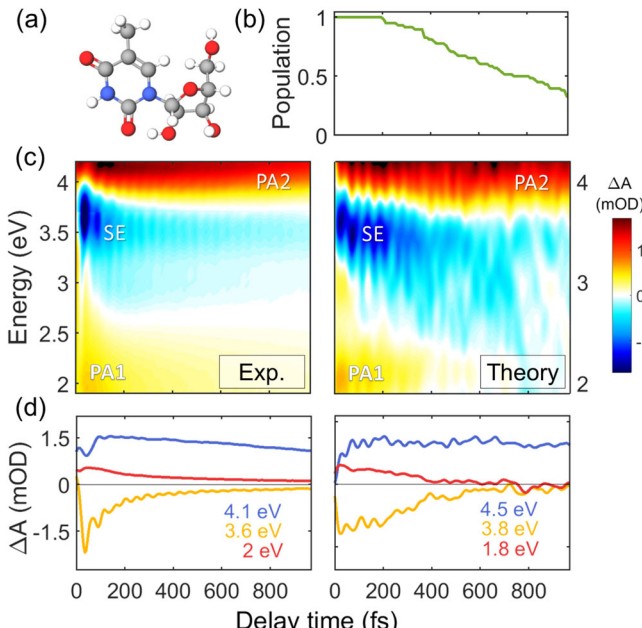

**Fig. 2 Excited state dynamics of 5-methyluridine. a** Molecular structure of 5mUrd. **b** Time-dependent excited-state population calculated by QM/MM simulations. **c** Experimental (left) and computed (right) TA maps of 5mUrd. Two photo-induced absorption bands (labeled PA1 and PA2) and a stimulated emission (SE) band can be identified. **d** TA dynamics at selected probe photon energies; signal intensity reported as differential optical density ΔA in mOD units. Source data are provided as a Source Data file.

photoinduced absorption (PA) feature below 2.4 eV (PA₁). We note that the rise times of the signals are affected by the instrumental response function and the presence of the coherent artifact[30–32] in the first 50 fs, which was minimized by using low pump fluences. An additional PA band can be observed at early times at 3.10 eV for orthogonal pump and probe polarizations (Supplementary Fig. S37), which is covered by the SE for parallel polarizations (Fig. 1c). Both the SE and PA₁ bands decay very quickly and completely vanish after ~200 fs, giving rise to another PA band (PA₂) at longer times covering the entire near-UV spectral window (3.10–4.25 eV). A global analysis of the data (Fig. S43) reveals that the SE and the PA₁ band decay with 97 fs time constant. This value compares well with results of fluorescence up-conversion measurements[27], although it is shorter than those found in previous TA experiments[19,23] which might be limited by the temporal resolution (Table S2).

The TA maps of 5mUrd, shown in Fig. 2, are qualitatively similar to those of Urd. At early times, we observe a SE signal in the UV region, confined in the 3.35–4.00 eV window, as well as a PA band extending below 2.4 eV (PA₁) and a second one above 4 eV (PA₂). As in Urd, an additional PA band is observed at 3.10 eV for crossed pump and probe polarizations (Supplementary Fig. S39). The key difference with respect to Urd is in the spectral dynamics of these features. The SE exhibits a clear red-shift during the first 200 fs accompanied by a significant broadening (see TA spectra in Supplementary Fig. S33) and survives up to 1 ps. Global analysis of the SE and PA₁ bands reveals time constants of 100 and 575 fs (Fig. S45). Buchner et al.[20], Xue et al.[24], Kwok et al.[33] and Gustavsson and co-workers[8,18,27] found two time constants (70–200 fs and 390 fs - 2.2 ps); in contrast, other works have described a mono[6,7,22] or biexponential decay with picosecond time constants[17,25] (Table S1). On the other hand, the PA₂ band exhibits a delayed build-up in the first 100 fs which correlates to the decay dynamics

of the SE. This rise is due to the red-shift of the SE band that uncovers the underlying PA. Subsequently, it decays on a few ps time scale accompanied by a blue-shift.

**Uridine decays ballistically to the ground state through a ring puckering channel.** To interpret the observed ultrafast relaxation dynamics, we compare them to TA maps computed from first principles, which are analysed using the same fitting protocol (Supplementary Note 6) applied to the experimental data. We use a mixed quantum-classical dynamics approach in a QM/MM framework, that couples a multireference dynamically correlated description (CASPT2) of the pyrimidine nucleobase with an explicit classical atomistic model (AMBER force field) of the sugar substituents and solvent. Further details are provided in Supplementary Note 2.

Figure 1c reports the computed TA spectra of Urd, which are in very good agreement with experiment. We assign the SE and PA₁ signals at early times to the ¹ππ* state (Supplementary Note 6.2). PA₂ is identified as PA from the hot ground state as the build-up time of this band matches the decay of the SE. This assignment is fully corroborated by the simulation and global analysis of the spectroscopic signals (Fig. S42 and Fig. S43). By fitting the population dynamics of the ¹ππ* state (Fig. 1b) with a mono-exponential function, we obtain a lifetime of 120 fs which compares well to the lifetime obtained by global fitting both the experimental (97 fs) and the modeled (110 fs) spectral dynamics of the PA₁ band.

Through static computations within the hybrid QM(CASPT2)/MM(AMBER) framework we recently identified two deactivation channels from the lowest bright ¹ππ* state to the ground state in pyrimidine nucleosides, involving ring-puckering with H₉ out-of-plane bending and N₁-C₂ bond breaking[34]. Both deactivation pathways are associated with CIs reached over small barriers (0.1–0.2 eV), making them potentially responsible for the ballistic ~100 fs decay observed experimentally. Our mixed quantum-classical dynamics simulations enable us to identify ring puckering[35] as the largely dominant excited state deactivation mechanism in Urd with excitation at 4.5 eV (Supplementary Note 3.6.1), though the limited number of trajectories do not eliminate the N₁-C₂ bond breaking as a possible, albeit minor, channel.

**Solvent reorganization and the inertia of the methyl group slow down the excited-state relaxation in 5-methyluridine.** The observed analogy in the overall appearance of the ultrafast TA spectra of Urd and 5mUrd is understood by the rather similar electronic structure of both nucleosides. This is also confirmed by the simulations of 5mUrd, which reproduce all relevant features (Fig. 2c) within the margin of error. The simulations recover the red-shift of the SE during the first 300 fs, which is accompanied by a rise of PA₂ above 4.00 eV. A global analysis of the "PA only" spectra assigns this signal to the tail of an intense PA band of the ¹ππ* state (Supplementary Fig. S41 and S48), revealed only upon red-shift of the overlapping SE signal. With time, the PA of the ¹ππ* state is progressively overlaid by the PA of the hot ground state, which decays in a few ps, thus rationalizing the apparent long lifetime of the PA₂ band in comparison to the remaining fingerprint bands of the ¹ππ* state. Two additional PA bands characterize the ¹ππ* state at early times, PA₁ below 2.10 eV, clearly visible in the simulated spectrum, and another one peaking at 3.30 eV, whose tail at 3 eV is faintly visible in the spectrum during the first 100 fs but it is clearly resolved in cross-polarized spectra (Supplementary Fig. S39). Upon red-shift the SE overlaps with both PA bands, leading to the ultrafast disappearance of the band at 3.30 eV and to pronounced intensity

fluctuations in $PA_1$, which nevertheless survives until the end of the simulation time window.

The non-adiabatic dynamics underlying the simulated spectra reveals a surprising picture. The lowest $^1\pi\pi^*$ state of 5mUrd is found to decay on a time scale of several hundred fs (a mono-exponential fit of the population dynamics in Fig. 2b reveals a lifetime of 750 fs), which is almost one order of magnitude longer than Urd. Moreover, only one hopping event is encountered during the first 200 fs. Thus, the fast component observed in the TA spectra (Fig. 2c) is not associated with the decay to the ground state.

Through mapping of the $^1\pi\pi^*$ potential energy surface[36] (PES) implemented on the entire ensemble of trajectories (Supplementary Note 7) we find that the solvent slows down ultrafast bulky geometrical deformations by introducing a dynamic barrier. This barrier adds up to the inertia of the methyl group, thus making it impossible for 5mUrd to reach the CI region on a sub-100 fs time scale. This analysis reveals that the population is dynamically trapped in the $^1\pi\pi^*$ state on the time scale of several hundred femtoseconds during which the solvent adapts to the electronic structure and the barrier decreases. The red-shift of the SE observed in both experimental and theoretical spectra represents a clear spectroscopic signature of the $^1\pi\pi^*$ trapping and of the coupled solvent-solute relaxation, whereas the accompanying signal spreading is shown to be a consequence of the pronounced methyl out-of-plane distortion in several snapshots. In contrast, no solvent reorganization is necessary for Urd due to the space-conserving deformation (involving only hydrogen atoms) towards the CI seam[36]. This facilitates the ultrafast decay on a sub-100 fs time scale.

**Internal conversion mechanism of 5-methyluridine.** After 1 ps about 20% of the trajectories in 5mUrd are found to still roam in the excited state PES. As for Urd, ring-puckering is found to be the dominant deactivation pathway, channeling about 90% of the decaying population; however, as mentioned above, in 5mUrd the puckering mode involves the methyl group (Supplementary Note 3.6.2). A few trajectories are found to explore $O_8$-out-of-plane bending during deactivation and a single trajectory (out of a total of 57) undergoes $N_1$-$C_2$ bond breaking (Supplementary Note 3.6.2). We note that, while this channel has been already identified in the literature, its potential involvement could only be hypothesized based on static PES analysis. Our dynamic simulations show that the ring-opening is a viable, albeit minor, decay channel. The longer excited state lifespan of 5mUrd facilitates the activation of diverse conformational degrees of freedom and it is thus allowed to explore various deactivation routes, which might render it potentially more prone to photodamage.

**Involvement of the $n\pi^*$ state in the deactivation of uridine and 5-methyluridine.** Could an ultrafast non-adiabatic transfer to a non-emitting excited state (e.g. $^1n\pi^*$) be held liable for the observed 100 fs time constant in 5mUrd? We thoroughly investigated this possibility by re-computing the energetics along all trajectories, this time taking into consideration further $^1\pi\pi^*$ and (the lowest) $^1n\pi^*$ states. Ad hoc non-adiabatic Tully fewest switches surface hopping dynamics in this new basis estimated that, even if 5mUrd spends a picosecond in the $^1\pi\pi^*$ state, it does not undergo effective non-adiabatic transfer to another excited state before reaching the CI seam with the ground state (Supplementary Note 3.5.2). A global analysis of the experimental data in the picosecond domain (Fig. S45) reveals a low long-living signal, which is presumably attributed to the $^1n\pi^*$ and $^3\pi\pi^*$ states. We estimate that ~5% of the population relaxes through this minor decay channel.

Somewhat surprisingly and in contrast with 5mUrd, we find that Urd is susceptible to a non-adiabatic transfer to the nearest $^1n\pi^*$ state, despite its short $^1\pi\pi^*$ lifetime. This behavior is attributed to the greater (with respect to 5mUrd) energetic proximity of the $^1\pi\pi^*$ and the $^1n\pi^*$ states in the FC region (Supplementary Note 3.2.3) leading more often to wavefunction mixing along the dynamics (Supplementary Note 3.5.1). In Urd we estimate an upper limit of 20% for the yield of non-adiabatic transfer from the $^1\pi\pi^*$ state to the $^1n\pi^*$ state. This is experimentally supported by the global analysis of the long living signal detected at ~3.50 eV (Supplementary Figs. S40, S41 and S47), also reported recently by Crespo and coworkers[23], and compatible with the calculated absorption signal of the $^1n\pi^*$ state at ~3.10 eV in our previously reported static calculations[37]. Both our analysis (Supplementary Note 6.2) and a recent study in literature[24] suggest that the fraction of the population that ends up in the $^1n\pi^*$ state continues its evolution towards the triplet state manifold via an ISC on the nanosecond time scale.

**Vibrational modes controlling the excited state dynamics of uridine and 5-methyluridine: signatures of the motion towards the CI.** The sub-30-fs temporal resolution of our experimental setup reveals impulsively excited coherent oscillations in the SE signal (Figs. 1c and 2c), thereby allowing an unprecedented insight into the excited state vibrational dynamics of the nucleosides. By applying a high bandpass Fourier filter, we obtain the oscillatory component of the TA spectra (time traces at different probe photon energies shown in Fig. 3a). In both systems the oscillations are initiated immediately with the excitation. In Urd they decay on the 100-fs timescale (similarly to the SE) and exhibit only a few periods, strongly supporting the excited state origin of the underlying vibrational dynamics. In 5mUrd, on the other hand, the coherent oscillations can be clearly resolved until 700 fs, in agreement with the much longer lifetime of the $^1\pi\pi^*$ state.

A 2D Fourier analysis of the oscillatory component of the simulated SE signals (Fig. 3b, lower panels) accurately reproduces the experimental 2D maps (Fig. 3b, upper panels). The experimental maps of both Urd and 5mUrd reveal a dominant mode with a frequency of 750 cm$^{-1}$. The presence of a node around 3.80 eV, which coincides with the maximum of the SE band, and the observed $\approx\pi$ phase shift between oscillations at probe photon energies to the red and the blue of this node, support the assignment of the coherence to the excited state[38]. This mode is clearly reproduced by the simulation, albeit at slightly lower frequency (~720 cm$^{-1}$); the same occurs for the node between 3.8 and 4.0 eV which closely matches the vertical energy gap with the ground state at the $^1\pi\pi^*$ excited state minima of both nucleosides[35]. We also note the weak mode below 600 cm$^{-1}$ present in the Urd map, whose amplitude is slightly overestimated in the simulation.

Excited state normal mode analysis (NMA, see Supplementary Note 3.7 for more details) allows identifying the modes responsible for the coherent oscillations in the SE. In both systems, the 750 cm$^{-1}$ mode is a breathing mode of the aromatic ring, characterized by a large amplitude of the $N_1C_6C_5$ angle (see insets in Fig. 3b). The ring breathing mode, together with high frequency $C_5C_6$ and CO stretching modes, which are beyond the limits of the temporal resolution of our experimental setup, dominate the excited state vibrational dynamics. The NMA demonstrates that in Urd photoexcitation delivers significant amount of vibrational energy in modes involving the $C_5$ and $C_6$ carbon atoms, eventually weakening the double bond and facilitating the ring puckering. NMA of Urd associates a frequency of 600 cm$^{-1}$ to the hydrogen out-of-plane bending accompanying the puckering (see inset of Fig. 3b). Thus, we

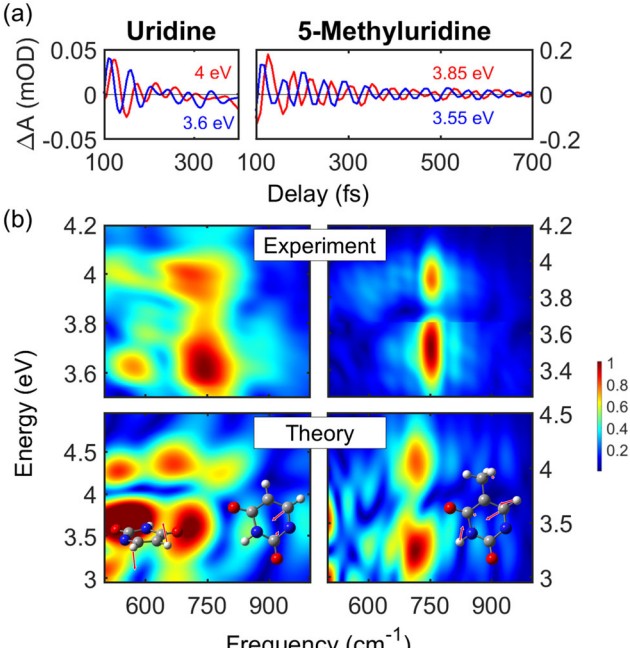

**Fig. 3 Coherent vibrations in pyrimidines. a** Oscillatory patterns were obtained after subtraction of the slowly varying component from the transient spectra maps by employing a bandpass Fourier filter. Two examples of the extracted oscillatory dynamics are shown for Urd (left panel) and 5mUrd (right panel). Signal intensity reported as differential optical density ΔA in mOD units. **b** Comparison of the experimental and theoretical two-dimensional Fourier transform maps of the residuals (oscillations) for Urd (left panels) and 5mUrd (right panels; the color jump in the experimental map is due to the merge of the two measurements corresponding to two different probe spectral regions). Inset left panel: schemes of the 600 cm$^{-1}$ (left) and 750 cm$^{-1}$ (right) vibrational modes in Urd. Inset right panel: scheme of the 750 cm$^{-1}$ vibrational mode in 5mUrd. Source data are provided as a Source Data file.

assign the peak below 600 cm$^{-1}$ observed in the experimental and theoretical Fourier transform maps of Urd to a signature of the coherent wave packet motion towards the CI seam (Fig. 3b).

As demonstrated in the analysis of the PES profile of the ensemble, in Urd a small barrier towards the CI exists at early times. We propose that detection of hydrogen out-of-plane puckering mode oscillations is a signature of this dynamical barrier, which disappears quickly, i.e., within 100–200 fs, yet allowing for a few oscillations (the mode has a period of ca. 55 fs). The lack of this signature in 5mUrd can be rationalized with the damping that the strongly repulsive potential of the surrounding water exercises on the methyl out-of-plane puckering, so that its amplitude is too small to cause a notable modulation of the SE signal.

To summarize, through the unprecedented combination of TA spectroscopy with sub-30-fs time resolution in the UV/visible and QM/MM simulations with highly correlated QM methods and explicit computation of spectroscopic signals, we derive a comprehensive picture of the decay mechanisms of water-solvated Urd and 5mUrd. We conclude that: (i) the sub-100-fs time constant dominating the spectral dynamics of Urd is associated with a coherent ballistic wave packet motion towards the $^1\pi\pi^*$/S$_0$ CI (Fig. 4a); (ii) the two-component decay for 5mUrd is to be ascribed to the motion away from the FC point (fast sub-ps component) and the subsequent solvent-assisted IC to the ground state (slow sub-ps component); (iii) the population of the dark $^1n\pi^*$ state is a minor decay channel and it is not associated with the slower sub-ps lifetime of 5mUrd. We identify ring puckering due to hydrogen (Urd) or methyl (5mUrd) out-of-plane bending as the main mechanism driving the evolution towards the CI. The longer excited lifetime of 5mUrd enables the activation of specific reaction coordinates (Fig. 4b).

We rationalize the difference in decay times with a solvent-induced dynamic barrier associated with the bulky methyl group motion. This postulates a novel concept of dynamical trapping in the $^1\pi\pi^*$ state of pyrimidine nucleobases. Our study discerns the twofold effect of the solvent: (a) electrostatic, i.e., destabilization

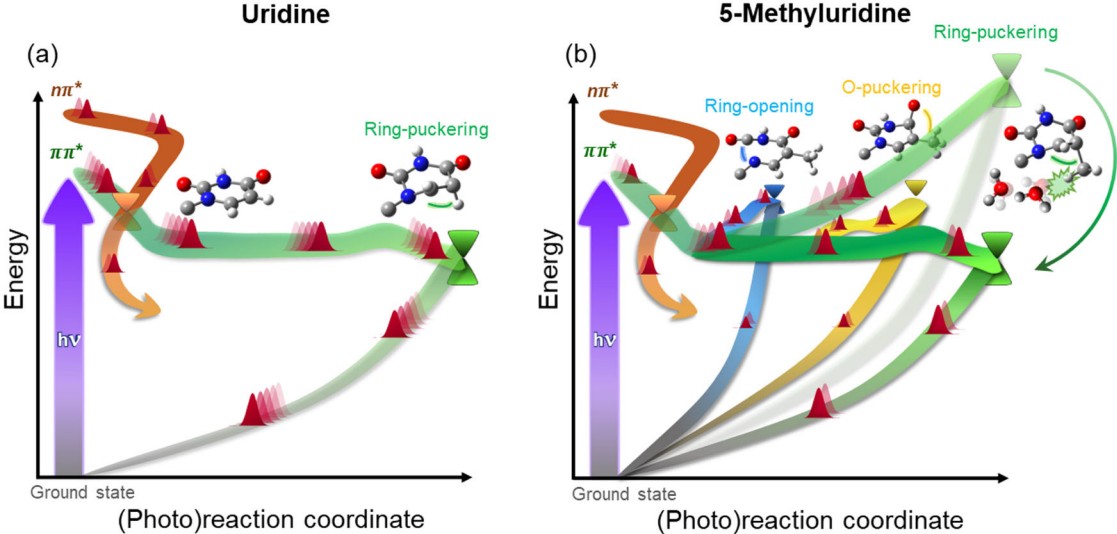

**Fig. 4 Excited state decay pathways in pyrimidines.** Schematic representations of photoinduced events in (**a**) Urd and (**b**) 5mUrd at sub-100 fs (semi-transparent surfaces) and sub-ps (opaque surfaces) times. In Urd, the bright $^1\pi\pi^*$ state (green surface) is depopulated on a sub-100 fs time scale through ring-puckering. 5mUrd is subject to a dynamical barrier due to the steric interaction with the solvent which adds up to the increased inertia of the methyl group and impedes decay on sub-100 fs time scale. Solvent-solute relaxation dynamics lowers the barrier on the time scale of few hundred fs allowing for a quasi-barrierless decay through ring-puckering as in Urd. The delayed decay facilitates energy redistribution and allows 5mUrd access to further deactivation pathways such as ring-opening and oxygen puckering. $^1n\pi^*$ population is found to be a secondary channel in both Urd and 5mUrd with estimated yields of ca. 20% and 5%, respectively.

of the $^1n\pi^*$ state which makes it less relevant for the non-adiabatic decay with respect to gas-phase; (b) steric, i.e., the slowing down of bulky geometrical deformations, which proceed unhindered in gas-phase. Eventually, we identify the experimentally detected 600 cm$^{-1}$ coherent vibration as the signature of the H-out-of-plane mode accompanying the ring puckering in Urd during the motion towards the CI.

## Methods

**Experimental setup**. Ultrafast TA experiments were carried out on a home-made pump-probe setup[28], based on a Ti:Sapphire laser (Libra, Coherent) delivering 100-fs pulses at 1.55 eV photon energy and 1 kHz repetition rate. A fraction of the laser power was used to feed a broadband visible non-collinear optical parametric amplifier (NOPA). The output pulses (1.77–2.38 eV), compressed to sub-10-fs duration by chirped dielectric mirrors, were successively frequency doubled in a 20-μm-thick Type I β-barium borate crystal, generating broadband UV pump pulses with spectrum spanning 4.43–4.6 eV. The UV pulses were compressed with a MgF$_2$ prism pair to nearly transform-limited 18-fs duration, characterized by two-dimensional spectral interferometry[29]. Broadband probe pulses, covering 1.9–3.9 and 3.5–4.6 eV, were obtained through white light continuum generation by focusing either the laser fundamental or its second harmonic in a slowly moving 2 mm-thick CaF$_2$ plate. The instrumental response function of the setup is estimated to be 25–35 fs, depending on the probe wavelength.

In order to avoid photodamage of the sample and generation of solvated electrons from water, the pump energy was limited to 20–30 nJ (resulting in a fluence of 88–132 μJ/cm$^2$) and a laminar flowing jet configuration (~0.15 mm path length) was employed. TA spectra of the pure solvent are provided in the Supplementary Information (Figs. S34 and S35). After the sample, the transmitted probe was sent to a spectrometer (SP2150 Acton, Princeton Instruments) and detected using a linear image sensor driven by a custom-built electronic board (Stresing Entwicklungsburo GmbH). For each probe wavelength, the differential absorption (ΔA) was measured as a function of the pump-probe delay. Measurements were recorded in parallel (main text), magic and orthogonal (Supplementary Note 5) pulse polarizations.

**Sample preparation**. 5-methyluridine (97% purity) and uridine (99% purity) were purchased from Sigma-Aldrich and used as received. A phosphate-buffered saline (PBS) solution was prepared by dissolving 3.6 g of sodium dihydrogen phosphate and 4.26 g of sodium hydrogen phosphate in ultrapure water to obtain a pH 7.4 and a concentration of 15 mM. The 5mUrd and Urd in PBS solutions were prepared to obtain concentrations respectively of 24.2 and 27.6 mM, resulting in an absorbance of 3 OD at the central pump wavelength. The steady-state absorption spectra are reported in the Supplementary Fig. S32.

**Computational methods**. Molecular dynamics simulations following Newton's equations of motion for the nuclei and utilizing hybrid QM/MM numerical gradients were performed at the full-π SS-2-CASPT2/SA-2-CASSCF(10,8) level of theory for 500 fs (Urd) and for 1000 fs (5mUrd) with a time step of 1.0 fs applying Tully's fewest switches surface hopping algorithm[39] with the Tully–Hammes–Schiffer (THS) scheme[40] and a decoherence correction as implemented by Persico et al.[41] (Supplementary Note 2.5). The state averaging covered the ground and the lowest $^1\pi\pi^*$ state. A High/Medium/Low Layer (HL/ML/LL) partitioning was applied to a spherical droplet centered at the nucleoside with a radius of 12 Å. The HL (QM region) comprises the nucleobase. The sugar and water molecules in 5 Å distance from the center of mass of the nucleoside were included in the movable ML. The remaining water molecules were kept fixed in the LL. The generally contracted basis set ANO-L adopting valence double-ξ contractions was utilized[42].

The simulations were performed for 57 geometry realizations (per system) selected out of 500 snapshots generated via Wigner sampling (Supplementary Notes 2.3 and 3.1) on top of a representative geometry taken from a classical molecular mechanics simulation (Supplementary Note 2.1) and refined within the QM/MM framework at the Möller-Plesset second order perturbation theory (MP2, Supplementary Note 2.2). The geometry optimizations and simulations were conducted with the COBRAMM program[43], interfacing the QM software Molcas 8[44] with the AMBER suite of classical force fields[45]. Wigner sampling was realized with a stand-alone script, part of the program JADE[46]. On the basis of the simulations, TA spectra were generated (Supplementary Note 2.7) after computing the excited state electronic structure (energies and transition dipole moments) of Urd and 5mUrd at the SS-20-CASPT2/SA-20-CASSCF(10,8) level of theory at every time step along the trajectories.

We performed an extensive methodology benchmarking on Urd concerning the active space size and the choice of CASPT2 flavor (see Supplementary Notes 2.5, 3.3 and 3.4). In particular, benchmarking of the single state (SS) flavor considering only the lowest $^1\pi\pi^*$ state (i.e. SS-2-CASPT2/SA-2-CASSCF(10,8), the level chosen for the production runs) against extended multi-state (XMS) which considers the lowest four $^1n\pi^*$ and $^1\pi\pi^*$ states (i.e. XMS-9-CASPT2/SA-9-CASSCF(14,10)) demonstrates an outstanding agreement between the two variants (see Supplementary Notes 3.4) validating the SS-CASPT2 level and strongly

supporting the predicted secondary role of the $^1n\pi^*$ state in the deactivation process. Due to their high computational cost, the XMS-CASPT2 simulations could be run only on a subset of 25 Urd trajectories, thus preventing collecting enough statistics to quantify the involvement of the $^1n\pi^*$ state. Consequently, its involvement in the internal conversion of Urd and 5mUrd was quantified by other means: (a) analysis of the extent of $^1n\pi^*/^1n\pi^*$ mixing in the FC region; (b) analysis of the evolution of the energy gap between the lowest $^1\pi\pi^*$ state and higher lying excited states along the computed trajectories; (c) ad-hoc non-adiabatic Tully fewest switches surface hopping dynamics in a diabatic basis in order to obtain an upper limit for the percentage of trajectories that could potentially depart from the $^1\pi\pi^*$ state before internal conversion to ground state (Supplementary Notes 2.6 and 3.5).

## Data availability

Data generated or analyzed during this study have been deposited in the Zenodo database (https://doi.org/10.5281/zenodo.5710891). Source data for Fig. 1–3 are provided in the Source Data file. Source data are provided with this paper.

## Code availability

The COBRAMM code used to perform the mixed quantum-classical simulations is available free of charge on Gitlab (https://gitlab.com/cobrammgroup/cobramm.git). The post-processing and analysis codes used to generate the spectra presented in this study are available from the corresponding authors upon request.

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

## Acknowledgements

We thank Dr. Stefano Santabarbara for preparing the PBS. P.K., G.C., and M.G. acknowledge support from the H2020 Grant Agreement number 765266 (Light-DyNAmics). G.C. and M.G. acknowledge support from the European Research Council Advanced Grant STRATUS (ERC-2011-AdG No. 291198) and R.B.V. from the Marie Curie Actions (grant no. 328110). M.G. acknowledges support from PRIN 2017 project PHANTOMS, Prot. 2017A4XRCA. I.R. and M.G. acknowledge the Agence National de la Recherche project FEMTO-2DNA (ANR-15-CE-29-0010). We acknowledge the use of HPC resources of the "Pôle Scientifique de Modélisation Numérique" at the ENS-Lyon, France.

## Author contributions

R.B.V., A.O., P.K., and C.M. built the experimental setup. P.K., R.B.V., A.O., L.G., and I.D. performed the measurements. A.N. and O.W. implemented the non-adiabatic dynamics algorithm. AN performed the simulations. R.B.V., A.N., P.K., I.C., and I.R. analysed the data. G.C. and M.G. conceived the idea and supervised the project. R.B.V. and A.N. wrote the manuscript with inputs from all the authors.

## Competing interests

The authors declare no competing interests.
