## [Peer Review File · Nature Communications]

Editorial Note: Parts of this peer review file have been redacted as indicated to maintain the confidentiality of other journals.

REVIEWER COMMENTS

Reviewer #1 (Remarks to the Author):

This is a magnificent paper combining the very best experimental and theoretical methods to understand a fundamental problem in DNA photophysics. This approach sets a very high benchmark for future studies exploring the excited state dynamics of nucleic acids.

The authors have answered all of my queries succinctly and appropriately. I therefore recommend publication of the manuscript without equivocation or hesitation in Nature Communications.

Reviewer #2 (Remarks to the Author):

The authors have made great effort to take into account all the comments by the reviewers and improve the paper. They have made some very nice improvements. The most important development in this new revision is the clarification of why thymidine decays slower than uridine, which is attributed to a dynamical barrier induced by solvent rearrangement. I find this to be a really novel concept that could have broader implications. I think that this is a main conclusion that justifies publication at this journal, and should be emphasized even more.

There are some other points that the authors have made in order to justify the novelty of their work that I would like to comment further.

A main argument the authors make is that previous studies have not reached a consensus in the mechanisms for decay of these molecules and the current work will resolve all disagreements and there will be no ambiguity of the assignments any more. Of course, reaching that point is very difficult, and I doubt that this will be the case here. I will again focus on the theory (which is what I am most familiar technically). It is critical that readers are convinced that the current SS-CASPT2 approach which only includes the pi^* excited state is sufficient to describe the effect of npi^* states, and the results are not an artifact of the methodology. For that reason I find it very surprising that the authors have XMS-CASPT2 results including npi^* states that validate the current approach, but they didn't include them in the current work because they want to save them for a future publication. The validation of the results with XMS-CASPT2 is a crucial part of making the current work one step closer to resolving the previous discrepancies. The authors can still publish a technical

paper later with more details, while including this crucial component here. In my opinion, including the XMS-CASPT2 comparisons is necessary for this work to be appropriate for publication to [redacted] (especially since these results are already available to the authors).

Another argument that is being made is the novelty of simulating the spectra directly.

On page 4 they say: "Moreover, spectroscopy simulations from first principles translating the molecular dynamics into experimentally observable spectral features have been completely missing so far." That is not true. There are numerous studies in the gas phase that have simulated the TRPES signal (starting even back in 2007 with the *J. Phys. Chem. A* 111, 8500-8508 (2007) paper, and very recently *J. Phys. Chem. Lett.* 2021, 12, 5099). Even in condensed phase, there are studies that somehow simulate the time resolved IR or other spectra (maybe not to the same extent and with the same beautiful 2D pictures reported here). For example, the Pilles et al. paper includes some simulated spectra.

Since a main conclusion of this work is the importance of the methyl inertia on the dynamics, the first study that described such an effect between uracil and thymine needs to be discussed here (*J. Chem. A* 111, 8500-8508 (2007)).

Minor points:

What is the physical reason that there is strong mixing in uracil between $\text{pipi}^*/\text{npi}^*$ but not in thymidine? This is observed already in the FC region, so it is not an effect of the slower dynamics in thymidine.

Decay to npi^* and triplet states may make the molecule more susceptible to photodamage. In that case, uracil may be more susceptible to damage based on the current results. In general, I agree with reviewer 3 that this extrapolation does not need/ or is not justified to be emphasized.

Page 15: "Excited state normal model analysis" should be "Excited state normal mode analysis"

Reviewer #3 (Remarks to the Author):

I have reviewed an earlier version of this manuscript for [redacted]. I believe the authors remarkably improved it in this new submission to Nature Communications and therefore I definitively recommend it for publication. I congratulate the authors for the seriousness with which they faced the Referees criticisms. Well done!

As I side note, I confirm my opinion that the photophysics of T and U was substantially already known. Of course, one can probably always find a paper where an opposite thesis is valued, and the authors show some of them, but I keep on thinking that people who worked for long time on these systems nowadays agree on the main facts.

Nonetheless I believe that things are now more evident with these new data and, for sure, in this revised version the authors highlighted much better a number of truly interesting facts on the solvent-induced dynamical barrier and on vibrational coherences.

Very minor notes:

Page 11, lines 7-8 "Thus, it isstate" I've the feeling this sentence can be read with two opposite meanings. I suggest the authors to rephrase it.

Page 12 " a single trajectory". The fact that it is just 1 trajectory it is not very informative until one reads at the end of the manuscripts that 60 trajectories were run.

Page 12. The authors choose to name Thymine "5mU": ok this is fine. At page 12 however after they used 5mU already several times, they use once "thymine". Although of course this creates no confusion to people with some expertise on the subject, I'd suggest to clarify from the beginning of the manuscript that 5mU and Thymine are synonyms (or if they are not , why..)

Page 14 "vertical energy gap". I suggest to clarify "vertical energy gap with the ground state"

Page 15. In the caption of Fig 3 they authors write that in the left panel the mode at 600 cm^{-1} is sketched on the right and the one at 750 on the left. I believe this is a typo and the opposite is true.

Page 17. The authors state “ the population of $1n\pi^*$ is a minor decay channel and is not associated with the slower...”. It is not fully clear to me (and maybe the sentence is a bit ambiguous) if the authors intend to refer to the slower decay they measured, or they mean also to rule out the possibility that the decay >100 ps observed by Hare et al has to do with the formation (at early times) of the $n\pi^*$. More specifically in that paper the authors discuss a possible $n\pi^* \rightarrow S_0$ decay or also the possible formation of triplet states from the $n\pi^*$ one.

Page 17 “might by” should be “might be”

Reply to Reviewers

We thank the referees for their effort in reviewing the original manuscript and for their numerous valuable comments which stimulated us to improve the draft. We thank the three reviewers for unanimously endorsing the publication of a revised draft in Nature Communications.

Reviewer #1

This is a magnificent paper combining the very best experimental and theoretical methods to understand a fundamental problem in DNA photophysics. This approach sets a very high benchmark for future studies exploring the excited state dynamics of nucleic acids.

The authors have answered all of my queries succinctly and appropriately. I therefore recommend publication of the manuscript without equivocation or hesitation in Nature Communications.

We thank the reviewer for his/her positive appraisal of our work.

Reviewer #2

The authors have made great effort to take into account all the comments by the reviewers and improve the paper. They have made some very nice improvements. The most important development in this new revision is the clarification of why thymidine decays slower than uridine, which is attributed to a dynamical barrier induced by solvent rearrangement. I find this to be a really novel concept that could have broader implications. I think that this is a main conclusion that justifies publication at this journal, and should be emphasized even more.

We thank the reviewer for his/her suggestion. In order to stress more this point we have added the following sentences in the revised manuscript:

“Key questions regarding the assignment of the experimental decay times, the involvement of dark states, and the molecular processes responsible for the higher reactivity of thymine, remain open. *Furthermore the role of the solvent and how it affects the dynamics is still to be explored.*”

“ *These results provide a novel view of the decay process that calls for a key contribution of solvent reorganization, thereby disentangling two contributions: a static (destabilization of the $n\pi^*$ state with respect to the $\pi\pi^*$ state) and a dynamic (slowing down ultrafast bulky geometrical deformations) one.*”

To emphasize our conclusion even more, we have changed the title of the section “The origin of the fast sub-picosecond component in the spectra of 5-methyluridine: relaxation from the Franck-Condon region” to “*Solvent reorganization and the inertia of the methyl group slow down the excited-state relaxation in 5-methyluridine*”.

There are some other points that the authors have made in order to justify the novelty of their work that I would like to comment further.

Referee's comment 1): A main argument the authors make is that previous studies have not reached a consensus in the mechanisms for decay of these molecules and the current work will resolve all disagreements and there will be no ambiguity of the assignments any more. Of course, reaching that point is very difficult, and I doubt that this will be the case here. I will again focus on the theory (which is what I am most familiar technically). It is critical that readers are convinced that the current SS-CASPT2 approach which only includes the $\text{p}\pi^*$ excited state is sufficient to describe the effect of $\text{n}\pi^*$ states, and the results are not an artifact of the methodology. For that reason I find it very surprising that the authors have XMS-CASPT2 results including $\text{n}\pi^*$ states that validate the current approach, but they didn't include them in the current work because they want to save them for a future publication. The validation of the results with XMS-CASPT2 is a crucial part of making the current work one step closer to resolving the previous discrepancies. The authors can still publish a technical paper later with more details, while including this crucial component here. In my opinion, including the XMS-CASPT2 comparisons is necessary for this work to be appropriate for publication to [redacted] (especially since these results are already available to the authors).

To address the comment from the referee we have removed the word “unambiguously” from the text of the manuscript. The new text now reads: “... *we are able to reproduce the experiments and ~~unambiguously~~ assign the observed spectral signals and the corresponding time constants to the specific pathways responsible for the decay.*” We agree with the referee that the draft would profit from documenting the XMS-CASPT2 calculations. In the revised version of the Supporting Information, we have now added a dedicated section comparing the three CASPT2 flavors SS, MS and XMS (Supplementary Note 3.4). This section extends a section in the previous version of the SI which concerned the comparison between the SS and MS flavors. Furthermore, some technical details on the MS and XMS trajectories have been added in the sec. 2.5 “Non-adiabatic mixed quantum-classical dynamics simulations” under subsection 2.5.2 “Technical details on the MS- and XMS-CASPT2 benchmarking dynamics on Uridine”.

Furthermore, the section **Methods** in the main manuscript was extended with a short discussion on the benchmarking:

“We performed an extensive methodology benchmarking on Urd concerning the active space size and the choice of CASPT2 flavor (see Supplementary Notes 2.5, 3.3 and 3.4). In particular, benchmarking of the single state (SS) flavor considering only the lowest $^1\pi\pi^$ state (i.e. SS-2-CASPT2/SA-2-CASSCF(10,8), the level chosen for the production runs) against extended multi-state (XMS) which considers the lowest four $^1n\pi^*$ and $^1\pi\pi^*$ states (i.e. XMS-9-CASPT2/SA-9-CASSCF(14,10)) demonstrates an outstanding agreement between the two variants (see Supplementary Note 3.4) validating the SS-CASPT2 level and strongly supporting the predicted secondary role of the $^1n\pi^*$ state in the deactivation process. Due to their high computational cost, the XMS-CASPT2 simulations could be run only on a subset of 25 Urd trajectories, thus preventing the collection of enough statistics to quantify the involvement of the $1n\pi^*$ state. Consequently, its involvement in the internal conversion of Urd and 5mUrd was quantified by other means: a) analysis of the extent of $^1\pi\pi^*/^1n\pi^*$ mixing in the FC region; b) analysis of the evolution of the energy gap between the lowest $^1\pi\pi^*$ state and higher lying excited states along the computed trajectories; c) ad-hoc non-adiabatic Tully fewest switches surface hopping*”

dynamics in a diabatic basis in order to obtain an upper limit for the percentage of trajectories that could potentially depart from the $^1\pi\pi^$ state before internal conversion to ground state (Supplementary Notes 2.6 and 3.5)."*

Referee's comment 2): Another argument that is being made is the novelty of simulating the spectra directly. On page 4 they say: "Moreover, spectroscopy simulations from first principles translating the molecular dynamics into experimentally observable spectral features have been completely missing so far." That is not true. There are numerous studies in the gas phase that have simulated the TRPES signal (starting even back in 2007 with the J. Phys. Chem. A 111, 8500-8508 (2007) paper, and very recently J. Phys. Chem. Lett. 2021, 12, 5099). Even in condensed phase, there are studies that somehow simulate the time resolved IR or other spectra (maybe not to the same extent and with the same beautiful 2D pictures reported here). For example, the Pilles et al. paper includes some simulated spectra.

We completely agree with the referee. In fact, the statement was not meant to be this general and was only referring to *UV-Vis transient absorption spectroscopy*. We are aware of existing simulations of TRPES and time-resolved IR. The sentence was modified as follows:

"Moreover, UV-Vis transient absorption spectroscopy simulations from first principles translating the molecular dynamics into experimentally observable spectral features have been missing so far."

Referee's comment 3): Since a main conclusion of this work is the importance of the methyl inertia on the dynamics, the first study that described such an effect between uracil and thymine needs to be discussed here (J. Chem. A 111, 8500-8508 (2007)).

Following the referee's suggestion, we included the citation to the paper:

"At odds with previous interpretations, our study shows that a decay to the ground state (GS) on the time scale of few hundred femtoseconds does not occur in 5mUrd due to a solvent-induced dynamic barrier associated with the bulky methyl group involved in the ring puckering. This effect adds to the already larger (with respect to Urd) inertia of 5mUrd due to the heavier methyl group, as recognized in earlier studies¹⁰."

Minor points:

Referee's comment 4): What is the physical reason that there is strong mixing in uracil between $\text{pipi}^*/\text{npi}^*$ but not in thymidine? This is observed already in the FC region, so it is not an effect of the slower dynamics in thymidine.

The reason for the stronger mixing between $\text{pipi}^*/\text{npi}^*$ in Urd with respect to 5mUrd is that the vertical energy gap between the pipi^* and npi^* states is somewhat smaller. The methyl group red-shifts the pipi^* state in 5mUrd by 0.1 eV (a similar observation has been reported previously for other methylated systems such as tryptophan compared to indole by Serrano-Andrés, L., B. O. Roos J. Am. Chem. Soc. 1996, 118, 185-195.). At the same time the npi^* state is not affected by the methylation. Thus, the pipi^* and npi^* states are energetically closer in Urd (on average 0.4 eV) compared to 5mUrd (on average 0.5 eV) and mix more often. The comparison of the $\text{pipi}^*/\text{npi}^*$

energy gaps at the FC point based on the Wigner sampling of 500 geometries in the FC region can be found in sec. 3.2.3 of the SI. To clarify this observation, we modified following sentence in the draft:

“Somewhat surprisingly and in contrast with 5mUrd, we find that Urd is susceptible to a non-adiabatic transfer to the nearest $1n\pi^$ state, despite its short $1\pi\pi^*$ lifetime. This behavior is attributed to the greater (with respect to 5mUrd) energetic proximity of the $1\pi\pi^*$ and the $1n\pi^*$ states in the FC region (Supplementary Note 3.2.3) leading more often to wavefunction mixing along the dynamics (Supplementary Note 3.5.1).”*

Referee’s comment 5): Decay to $n\pi^*$ and triplet states may make the molecule more susceptible to photodamage. In that case, uracil may be more susceptible to damage based on the current results. In general, I agree with reviewer 3 that this extrapolation does not need/ or is not justified to be emphasized.

As requested by the referees the paragraphs discussing the higher susceptibility to photodamage of 5mUrd have been removed.

Referee’s comment 6): Page 15: "Excited state normal model analysis" should be "Excited state normal mode analysis"

We thank the referee for spotting this typo.

Reviewer #3

I have reviewed an earlier version of this manuscript for [redacted]. I believe the authors remarkably improved it in this new submission to Nature Communications and therefore I definitively recommend it for publication. I congratulate the authors for the seriousness with which they faced the Referees criticisms. Well done!
As I side note, I confirm my opinion that the photophysics of T and U was substantially already known. Of course, one can probably always find a paper where an opposite thesis is valued, and the authors show some of them, but I keep on thinking that people who worked for long time on these systems nowadays agree on the main facts.

Nonetheless I believe that things are now more evident with these new data and, for sure, in this revised version the authors highlighted much better a number of truly interesting facts on the solvent-induced dynamical barrier and on vibrational coherences.

Very minor notes:

Page 11, lines 7-8 “Thus, it isstate” I’ve the feeling this sentence can be read with two opposite meanings. I suggest the authors to rephrase it.

We have rephrased the sentence in a clearer way:

“Thus, the fast component observed in the TA spectra (Fig. 2c) is not associated with the decay to the ground state.”

Page 12 “ a single trajectory”. The fact that it is just 1 trajectory it is not very informative until one reads at the end of the manuscripts that 60 trajectories were run.

We modified the sentence accordingly

“a single trajectory (out of a total of 60)”

Page 12. The authors choose to name Thymine “5mU”: ok this is fine. At page 12 however after they used 5mU already several times, they use once “thymine”. Although of course this creates no confusion to people with some expertise on the subject, I’d suggest to clarify from the beginning of the manuscript that 5mU and Thymine are synonyms (or if they are not , why..)

After removing the sentences associated with the higher photodamaging susceptibility of 5mUrd, as requested by referees #2 and #3, the naming confusion is no longer present.

Page 14 “vertical energy gap”. I suggest to clarify “vertical energy gap with the ground state”

The expression has been modified accordingly.

Page 15. In the caption of Fig 3 they authors write that in the left panel the mode at 600 cm⁻¹ is sketched on the right and the one at 750 on the left. I believe this is a typo and the opposite is true.

We thank the referee for spotting this typo.

Page 17. The authors state “ the population of 1npi* is a minor decay channel and is not associated with the slower...”. It is not fully clear to me (and maybe the sentence is a bit ambiguous) if the authors intend to refer to the slower decay they measured, or they mean also to rule out the possibility that the decay >100 ps observed by Hare et al has to do with the formation (at early times) of the npi*. More specifically in that paper the authors discuss a possible npi->S0 decay or also the possible formation of triplet states from the npi* one.

It was intended the slower (still sub-ps) decay in our measurements. The sentence has been modified to clarify the statement.

“the population of the dark 1npi state is a minor decay channel and it is not associated with the slower sub-ps lifetime of 5mUrd.”*

Page 17 “might by” should be “might be”

We thank the referee for spotting this typo.

REVIEWERS' COMMENTS

Reviewer #2 (Remarks to the Author):

The authors have addressed all the comments satisfactory and in great detail, so I fully support publication.

Reply to Reviewers

Reviewer #2

The authors have addressed all the comments satisfactory and in great detail, so I fully support publication.

We thank the reviewer assessing very positively our manuscript revisions and for supporting its publication.

We have also fully addressed the editorial requests and revised the manuscript to comply with the editorial policies and formatting requirements.